# Current-induced skyrmion generation and dynamics in symmetric bilayers

A. Hrabec[1,2], J. Sampaio[1], M. Belmeguenai[3], I. Gross[2,4], R. Weil[1], S.M. Chérif[3], A. Stashkevich[3,5], V. Jacques[2], A. Thiaville[1] & S. Rohart[1]

Magnetic skyrmions are quasiparticle-like textures which are topologically different from other states. Their discovery in systems with broken inversion symmetry sparked the search for materials containing such magnetic phase at room temperature. Their topological properties combined with the chirality-related spin–orbit torques make them interesting objects to control the magnetization at nanoscale. Here we show that a pair of coupled skyrmions of opposite chiralities can be stabilized in a symmetric magnetic bilayer system by combining Dzyaloshinskii–Moriya interaction (DMI) and dipolar coupling effects. This opens a path for skyrmion stabilization with lower DMI. We demonstrate in a device with asymmetric electrodes that such skyrmions can be independently written and shifted by electric current at large velocities. The skyrmionic nature of the observed quasiparticles is confirmed by the gyrotropic force. These results set the ground for emerging spintronic technologies where issues concerning skyrmion stability, nucleation and propagation are paramount.

[1] Laboratoire de Physique des Solides, Univ. Paris-Sud, Université Paris-Saclay, CNRS, UMR 8502, F-91405 Orsay Cedex, France. [2] Laboratoire Charles Coulomb, Université de Montpellier and CNRS UMR 5221, 34095 Montpellier, France. [3] LSPM (CNRS-UPR 3407), Université Paris 13, Sorbonne Paris Cité, 99 avenue Jean-Baptiste Clément, 93430 Villetaneuse, France. [4] Laboratoire Aimé Cotton, CNRS, Université Paris-Sud, ENS Cachan, Université Paris-Saclay, 91405 Orsay Cedex, France. [5] International Laboratory MultiferrLab, ITMO University, St. Petersburg 197101, Russia. Correspondence and requests for materials should be addressed to S.R. (email: stanislas.rohart@u-psud.fr).

Over the past 2 years, a concerted effort has been made worldwide to study how magnetic skyrmions, a chiral phase evidenced in materials with broken inversion symmetry[1–4], can appear and be displaced in ultrathin ferromagnetic films and nanotracks[5,6]. From an experimental viewpoint, there remain three important challenges that are determinant for whether skyrmions will be useful in future data storage technologies. First, the ability to tailor the chirality and energy of domain walls (DWs), such that isolated skyrmions remain sufficiently stable at room temperature against the ferromagnetic ground state. Second, it is important to drive the skyrmions efficiently with spin-transport torques, such as the spin Hall effect (SHE), as the reproducible displacement of skyrmions is crucial to information transfer. Third, it is essential to be able to nucleate skyrmions readily, since the ability to write (new) information is primordial to any storage application. Experimental demonstrations of certain aspects of these three points have been reported[4,7–11], however, independent writing and shifting in a single device remains a challenge.

Recently, significant developments in spintronics have being directly connected with broken inversion symmetry. On one hand, this allows spin–orbit torques to improve the efficiency of current-induced magnetization manipulation[12–15]. On the other hand, such a situation allows to tailor DW chirality and lower the DW energy through the Dzyaloshinskii–Moriya interaction (DMI)[16–19], a requirement that ultimately permits the stabilization of skyrmions and their use in spintronics[2,3,5]. In general, the search for skyrmion host media requires a large DMI, which restricts the choice of materials.

Here we show that a globally symmetrical situation in magnetic bilayers meets all the requirements to host skyrmions, without the need for a very large DMI, since the control of DW chirality and energy is assisted by dipolar coupling. It results in two superimposed skyrmions, strongly coupled through their dipolar stray field, which behave as a single particle called skyrmion hereafter for simplicity. This method, compatible with spin–orbit torque-induced dynamics, offers a larger flexibility for the choice of materials. We show for the first time a simple and elegant technological way to independently write and shift such skyrmions at large velocities in a single functional device by means of electric current.

## Results

**Multilayer stack design.** Stabilization of isolated skyrmions requires a fine control of the DW energy[20] between two limiting cases: a large positive energy causes skyrmions to collapse, while a large negative wall energy destabilizes the collinear order and requires high magnetic fields to access the isolated skyrmions[7]. While this is generally achieved using DMI, we show that dipolar coupling in bilayers with perpendicular magnetization can be successfully used too. As illustrated in Fig. 1a on a magnetic bilayer, the stray field arising from the domains couples to the DW magnetization, in a flux-closure configuration, and promotes Néel walls with opposite chirality in both layers and thus lowers the DW energy[21]. The strength of this effect can be easily tuned by adjusting the spacer thickness, offering an additional means of control. The symmetric superimposition of two magnetic layers, each of them in a non-symmetric stacking, allows to satisfy both dipolar coupling and DMI. As the sense of the magnetization rotation induced by the stray field cannot be changed (left and right handed in the bottom and top layer, respectively), the large spin–orbit layers generating the interfacial DMI should be placed as spacer if the induced DMI parameter $D$ is positive, or as outer layers if it is negative. Another advantage of bilayers is an increase of the dipolar interaction between the skyrmion core and the

ferromagnetic surrounding as compared to a single layer, which further stabilizes skyrmions. As demonstrated below, this structure is also compatible with SHE-induced motion of skyrmions.

We use a stack of $Pt(5\,nm)\backslash FM\backslash Au(d)\backslash FM\backslash Pt(5\,nm)$ where $FM = Ni\backslash Co\backslash Ni$. The chosen FM gives the opportunity to change the surrounding metals without significantly changing the anisotropy which arises predominantly from the $Ni\backslash Co$ inter-faces[22]. Moreover, the use of two different ferromagnets gives more freedom to tune the stray field and the anisotropy by adjusting the Co and Ni thicknesses. Magnetometry and Brillouin light-scattering spectroscopy on single magnetic layers with both stacking orders has shown that both layers are similar, which guaranties film symmetry, only the sign of the DMI constant being opposite ($D = -0.21 \pm 0.01\,mJ\,m^{-2}$ for Pt/FM/Au and $D = +0.24 \pm 0.01\,mJ\,m^{-2}$ for Au/FM/Pt stacks, respectively), as expected (Supplementary Note 1). The DMI induced at the Pt/Ni interface thus yields a left-handed (counter-clockwise) chirality[23] as sketched in Fig. 1a, which justifies the use of Pt layers as bottom and top layers, in order to satisfy the dipolar interaction. The thickness of Pt has been chosen with respect to the reported spin-diffusion length to maximize the SHE[24] and Au remains a neutral layer due to negligible SHE[25] and small DMI[26]. While exchange coupling has been already explored theoretically[27] and experimentally[28] for stabilizing skyrmions, here we focus on dipolar coupling. The thickness of Au is therefore chosen to avoid interlayer exchange coupling[29] but also to minimize the electric current shunting (Supplementary Note 1).

**Skyrmion stabilization using dipolar couplings.** To understand the mechanism of the isolated skyrmion stabilization, one has to disentangle and quantify several energies involved in this process. The specific DW energy (hereafter called simply energy) $\sigma_0$, including the exchange and anisotropy energies[30], is lowered by $\pi D$ upon introduction of DMI[18,31]. The DMI favours a Néel wall structure, which also gives rise to magnetostatic charges on either side of the wall and creates a field opposed to the wall magnetization (red arrows in Fig. 1a)[31]. This is expressed by the energy increase $\delta_N$. Alongside these usual energy terms, in the case of the bilayer system one has to take into account additional energies. The DW–DW magnetostatic interaction between two DWs with opposite chiralities leads to another energy term $\delta_{DW-DW}$ which is illustrated by blue arrows in Fig. 1a. For spacer thicknesses $d$ smaller than the DW width $\Delta$, the magnetostatic charges created by each wall are so close that $|\delta_N| \approx |\delta_{DW-DW}|$ so that these two contributions almost compensate. On the contrary, the stray field arising from the domains, depicted by the green arrows in Fig. 1a gives another significant energy decrease, $\delta_{D-DW}$, which scales[32] approximately as $1/d$ and reinforces the chiral nature of each wall[21]. The DW energy within such system then reads

$$\sigma \cong \sigma_0 - \pi D - \delta_{D-DW}. \quad (1)$$

The quantitative analysis of the energies has been performed by micromagnetic calculations. Figure 1b shows the DW energy dependence on the spacer thickness. The DW energy increases with increasing $d$ as expected from the argument of $1/d$ dependence. However, when the spacer thickness $d$ is comparable with the DW width shown in the inset of Fig. 1b, the DW energy decreases with $d$. The DW width strongly responds to the spacer thickness to accommodate the important energy changes. One can define the optimum spacer thickness $d_{opt}$ where the effect of dipolar energy contribution is maximized. With our parameters, this analysis yields $d_{opt}$ close to 3 nm, and the ratio between the DMI and magnetostatic energies is $\delta_{D-DW}/\pi D = 1.1$, that is, half of the energy minimization is due

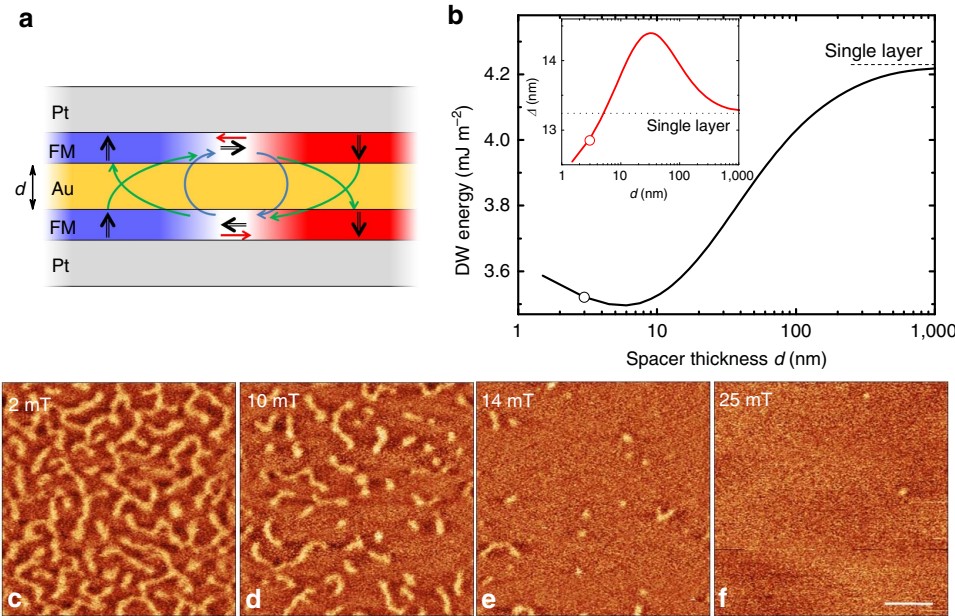

**Figure 1 | Skyrmion stabilization.** (**a**) Sketch of the Pt/FM/Au/FM/Pt stack. The black arrows indicate magnetization orientation inside the two layers containing a DW. The coloured arrows correspond to DW internal (red), DW–DW (blue) and domain–DW (green) magnetostatic interactions respectively. (**b**) Calculated DW energy and width parameter $\Delta$ as a function of the Au spacer thickness $d$. The circles correspond to the experimental case $d = 3$ nm. (**c**–**f**) Field-dependent MFM images revealing the process of skyrmion formation. The bright and dark contrast corresponds to repulsive and attractive force respectively. The sample is in a demagnetized state at low field. The remagnetization with increasing field leads to condensation of skyrmions and a decrease of their density. Scale bar, 1 μm.

to the magnetostatic energy. This approach based on isolated DWs can be also extended to 360° DWs and skyrmions where the stray field is slightly lower, showing that the dipolar mechanism is efficient to stabilize skyrmions down to at least 20 nm (Supplementary Note 2).

While the DW energy remains positive here, the dipolar coupling between the skyrmion core and the ferromagnetic surrounding can efficiently lower the skyrmion energy cost. However, in ultrathin films, dipolar coupling vanishes with the thickness[33] so that the film thickness must be larger than a characteristic length $l_c = \sigma / \mu_0 M_s^2$ to spontaneously promote magnetic textures[9,33] (with $M_s$ the spontaneous magnetization). In our case, $l_c = 3.9$ nm so that a single-layer film (1.5 nm thick) can hardly be demagnetized and shows a full remanence square hysteresis loop. On the contrary, bilayer films have a total thickness larger than $l_c$ and therefore spontaneously demagnetize in a multiple domain state, as shown by a zero remanence hysteresis loop (Supplementary Note 1).

The magnetic texture imaged by magnetic force microscopy (MFM, see Methods) confirms that the bilayers are at remanence in a worm-like demagnetized state. Figure 1c–f show a sequence of MFM images at different fields and illustrate how the worm-like structure can be unwound into the isolated skyrmion phase with moderate fields. As soon as an out-of-plane magnetic field is applied the domains start to contract into skyrmions whose size and density decrease with the magnetic field. Close to the saturation field only a few isolated skyrmions remain. They are the ones needed for free skyrmion dynamics studies and applications. We note that the MFM does not provide any insight into the topology of the created textures, a point addressed later. In the following, we focus on skyrmions in nanostructures. In those, size effects lower the magnetic field needed to saturate the sample, so that applying only 6 mT is sufficient to study isolated skyrmions of 160 ± 40 nm diameter, measured within the accuracy of the MFM (see Fig. 2).

**Skyrmion nucleation.** Several methods have been proposed to nucleate skyrmions[6–8,34–35]. Here we demonstrate another method which is characterized by its simplicity and integrability. To study the skyrmion dynamics in confined structures we have fabricated the device shown in Fig. 2a containing four parallel, 1 μm wide wires. The electrical contacts are fabricated in a non-symmetric fashion where one side of the wires is connected by sharp tips and the other by a wide electrode. At the tip current lines divergence[36], heating, and spin accumulation[37] are largest, disturbing the magnetic configuration. As a result, skyrmions spontaneously appear there and are injected into the track. Figure 2b shows a systematic skyrmion nucleation in two adjacent wires in the vicinity of the contact. Skyrmions are injected into fully saturated wires (−6 mT applied field) by a series of 7-ns-long pulses. Above a threshold of $j_c \simeq 2.6 \times 10^{11}$ A m$^{-2}$ the skyrmions are injected at the tip, as shown after application of the first pulse, and carried away into the tracks. When the polarity of the electric current is reversed this injection mechanism does not work, as the nucleated skyrmions are pushed by the current toward the tip. Nucleation at the wide electrode has never been observed. Therefore due to this geometrical asymmetry one polarity of the electric current serves as a skyrmion generator while the other simply shifts the existing skyrmions.

**Skyrmion dynamics.** After filling the tracks with skyrmions we reverse the polarity of the current to avoid interaction with newly injected skyrmions, and so switch to the skyrmion shifting mode. To measure the velocities as a function of current density, we measure the skyrmion displacement after application of each current pulse, of duration ranging from 3 to 10 ns. Figure 2c shows a sequence of images demonstrating skyrmion displacement by 3-ns-long pulses with a current density $j = 3.9 \times 10^{11}$ A m$^{-2}$. The resulting measured skyrmion velocity as a function of current densities at $B_z = -6$ mT presented in Fig. 2d reveals skyrmion velocities up to 60 m s$^{-1}$. From one

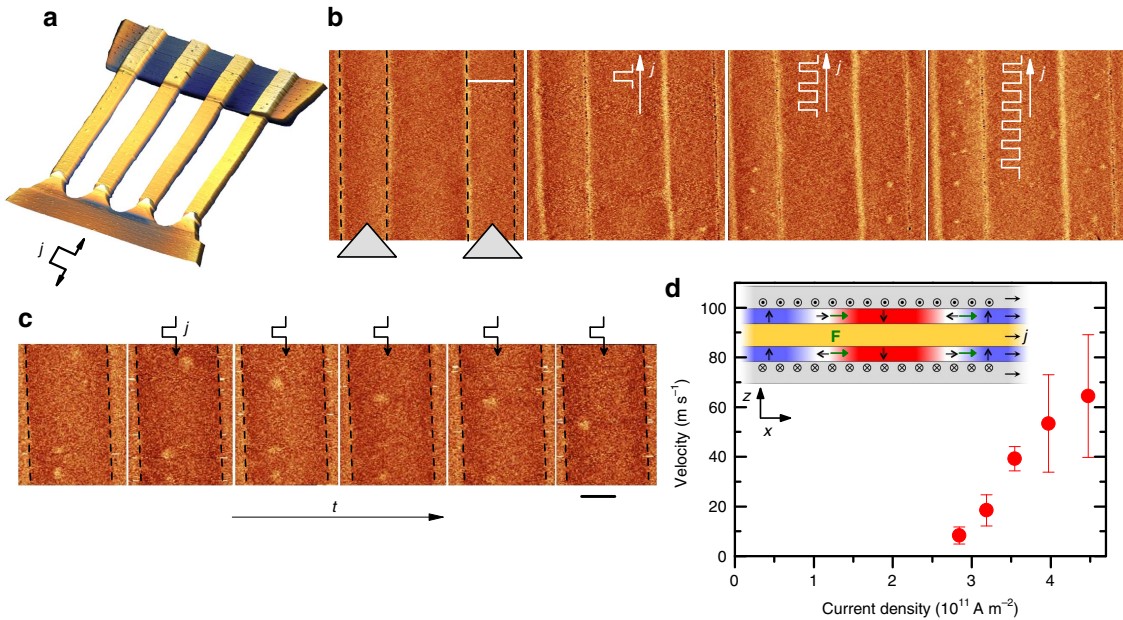

**Figure 2 | Skyrmion generation and dynamics. (a)** AFM image of the asymmetric device with the skyrmion injection tips. **(b)** Skyrmion writing by at $B_z = -6\,\text{mT}$ by 7 ns long, $j = 2.8 \times 10^{11}\,\text{A m}^{-2}$ pulses starting from a fully saturated state, with one injected skyrmion after application of first pulse, and several skyrmion after a train of pulses. The dashed lines and triangles correspond to the wire edges and electric contacts respectively. Scale bar, 1 μm. **(c)** Series of images showing skyrmion shift along the track between 3 ns, $j = 3.9 \times 10^{11}\,\text{A m}^{-2}$ electric pulses. Scale bar, 500 nm. **(d)** Measured velocity of skyrmions as a function of current densities at $B_z \simeq -6\,\text{mT}$. Error bars correspond to the s.d. The inset shows a sketch of the skyrmion cross section with the spin accumulation due to the electrical current j. The resulting force **F** (green arrows) defined by equation (2) causes both skyrmions to move in the same direction (against the electron flow). AFM, atomic force microscopy.

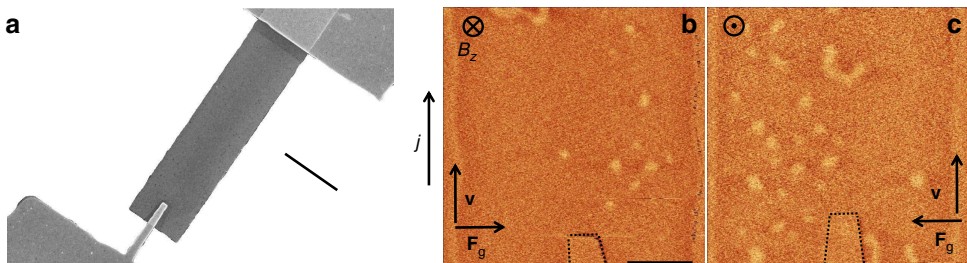

**Figure 3 | The gyrotropic force. (a)** SEM micrograph of the sample geometry used for skyrmion generation demonstration. The 3 μm ferromagnetic stripe is connected to Ti\Au pads with an injection finger 300 nm wide. Scale bar, 3 μm. **(b)** Resulting state after application of a series of 8 ns long pulses in an external field of $B_z \simeq -6\,\text{mT}$ where $j = 3 \times 10^{11}\,\text{A m}^{-2}$. Scale bar, 1 μm. **(c)** The same experiment with $B_z \simeq +6\,\text{mT}$ shows skyrmion accumulation on the left side of the device. The dashed line indicates position of the Ti/Au finger. SEM, scanning electron microscopy.

pulse to another, even at the largest current densities, successive displacement lengths are not equal, which underlines the role of defects and that skyrmions move by hopping within the potential landscape[38]. They advance in the track until they get pinned, or reach a strongly pinned skyrmion which prevents further propagation by skyrmion-skyrmion repulsion.

The skyrmions move in the direction opposite to the electrons which suggests that the spin Hall effect governs their dynamics[6,15]. When an electric current passes through the Pt layers, a spin accumulation with opposite polarities is generated at each interface, as sketched in Fig. 2d. The force acting on a skyrmion can be expressed as[6]

$$\mathbf{F}_{\text{SH}} = \pm \frac{\hbar}{2e} \pi j \theta_{\text{SH}} b \mathbf{e}_z \times \mathbf{e}_p \qquad (2)$$

where $j$ is the current density, $\theta_{\text{SH}}$ is the SHE angle and $b$ is a skyrmion characteristic length (half its perimeter when the skyrmion radius $R$ is much larger than the DW width parameter $\Delta$). The SHE-induced spin accumulation is along the vector $\mathbf{e}_p = \mathbf{n} \times \mathbf{j}$, with $\mathbf{n}$ being the outer normal to the SHE layer at the interface considered, and its sign is given by that of the SHE angle $\theta_{\text{SH}}$ (positive for Pt[24]). With $\mathbf{e}_z$ the vertical direction in the laboratory frame (for example, from substrate towards film), used to define the chirality of the skyrmions, the sign of the force is set by the chirality, as specified by the ± symbol in equation (2) where + stands for right-hand chirality. Since the spin accumulation and the chirality are both opposite at each interface, the forces acting on the skyrmions depicted in Fig. 2d point in the same direction in both magnetic layers. The skyrmions in both layers are therefore pushed in the same direction, along the electrical current here. The obtained velocities quantitatively agree with the model for free skyrmion dynamics in a disorder-free medium (Supplementary Note 4).

**Insight into topology via skew deflection.** One way to confirm that we deal with topological textures is to prove that they experience a gyrotropic force[39], expressed as

$$\mathbf{F}_{\text{G}} = \mathbf{G} \times \mathbf{v} = \left(0, 0, -\frac{\mu_0 M_s t}{\gamma_0}\Omega\right) \times (v_x, v_y, 0) \qquad (3)$$

where **G** is the gyrotropic vector, $\Omega = 4\pi Sp$ with $S$ being the winding number and $p$ the core polarity. $\Omega$ thus intimately binds the topology of the quasiparticle to its motion, providing a way to reveal its topological state. The skyrmion dynamics can be described by the massless Thiele equation[39,40]

$$\mathbf{G} \times \mathbf{v} - \alpha \mathcal{D} \cdot \mathbf{v} + \mathbf{F}_{\text{SH}} = 0 \qquad (4)$$

where $\mathbf{F}_{\text{SH}}$ is the force expressed by equation (2), $\alpha$ the Gilbert damping and $\mathcal{D}$ the dissipative tensor (Supplementary Note 4 for its calculation). The gyrotropic force therefore deviates the skyrmions from the direction of $\mathbf{F}_{\text{SH}}$ in a similar manner as electrons moving in a magnetic field[41–43].

To study this behaviour we have lifted the geometrical constrain and patterned the ferromagnetic stack into a $3\,\mu\text{m}$ wide strip where the skyrmions can move freely in the lateral direction. The electrical contact is designed asymmetrically by a 300 nm wide non-magnetic finger and a straight electrode as shown in Fig. 3a. We have verified that in this geometry the skyrmions can be only generated by one polarity of the current similarly to the case shown in Fig. 2b. The ferromagnet was first saturated and then a series of 8 ns long, $j = 3 \times 10^{11}\,\text{A}\,\text{m}^{-2}$ current pulses at $B_z = -6\,\text{mT}$ has been applied in order to inject the skyrmions. Figure 3b demonstrates a resulting magnetic configuration after application of a series of pulses showing several skyrmions. The skyrmions are ejected at the tip and are carried away by the current with a certain deflection towards right. Note that this deflection is also visible in Fig. 2c (to the left as the current is opposite). In order to prove the behaviour predicted by equation (3) we have generated skyrmions with opposite core magnetization by applying $B_z = +6\,\text{mT}$, that is, changing the sign of $\Omega$. Figure 3c shows a resulting skyrmion configuration after application of a train of pulses of the same polarity as in Fig. 3b. In this case, the skyrmions accumulate on the left side of the sample. We emphasize that the direction of the magnetic field shown in Fig. 3b,c is absolute and the skyrmion accumulation indeed appears on the side predicted by equation (3). The effect of Oersted field can be excluded as it would give the opposite deflection direction (Supplementary Note 3). This undoubtedly confirms that our textures have $S > 0$ topology compatible with skyrmions. A quantitative determination of the winding number is impossible as the track width and the skyrmion trajectory between the imperfections limit the deflection[42,44,45]. Note that $S = 1$ state would imply a deflection angle of $\approx 73°$ (Supplementary Note 4).

## Discussion

The guideline used here widens the possibilities of skyrmion stabilization and manipulation and, hence, their applications. Indeed, we have engineered a magnetic bilayer system which efficiently employs all the available energies (DMI and dipolar couplings) to stabilize a pair of skyrmions. The two skyrmions have the same topological charge while having opposite chiralities and are strongly coupled through their dipolar stray field. The opposite chirality in combination with a reversed spin accumulation results in a system suitable for the current-induced dynamics. We have developed functional devices with asymmetric electrodes designed for systematic current-induced skyrmion generation and motion. The skyrmion nature of the quasiparticles was demonstrated by topological filtering employing the gyrotropic force.

## Methods

**Sample preparation.** Samples were grown in an ultra-high vacuum evaporator with base pressure of $10^{-10}\,\text{mBar}$. The multilayers of Pt\FM\Au\FM\Pt with FM = Ni(4 Å)\Co(7 Å)\Ni(4 Å) were deposited on a high-resistive silicon with native oxide layer in order to minimize the Joule heating effect during electric current pulse application[46]. The symmetry of the stack has been verified by magnetometry measurements on Pt/FM/Au and Au/FM/Pt stacks using SQUID

($M_s = 0.85\,\text{MA}\,\text{m}^{-1}$, anisotropy field of about 150 mT), and Brillouin light-scattering spectroscopy (BLS) to determine DMI[47] (see Supplementary Note 1). The films were patterned by e-beam etching using an aluminium hard mask, which was consequently removed by chemical etching. The Ti\Au contacts were made in the second step via lift-off technique.

**Magnetization imaging and dynamics.** MFM was performed on a commercial Bruker Dimension 3000 microscope with a stage customized for high frequency transport measurements with a typical pulse rise/fall time $< 1\,\text{ns}$ (ref. 48). The MFM tips are home made with a non-magnetic capping layer in order to increase the distance between the magnetic tip coating and the surface to minimize the magnetic perturbations during the atomic force microscopy scan. Due to the fact that the tips are magnetically extremely soft and so follow the applied field, the skyrmions always appear as a bright (that is, repulsive) contrast due to the antiparallel magnetic configuration between the tip and the skyrmion. To avoid any heating and related thermal drifts, we have used a permanent magnet which implies an error of 20% on the given values of magnetic field. Current densities are defined as the average across the entire sample thickness (magnetic and non-magnetic layers).

**Micromagnetic calculations.** Micromagnetic modelling was carried out using the OOMMF code[49] where the cell size used was $0.5\,\text{nm} \times 1.5\,\text{nm} \times 1.5\,\text{nm}$. We used the measured material parameters $K_u = 0.5\,\text{MJ}\,\text{m}^{-3}$, $M_s = 0.85\,\text{MA}\,\text{m}^{-1}$ and used $A = 12\,\text{pJ}\,\text{m}^{-1}$ as an average of the bulk exchange constants for Co and Ni.

**Data availability.** All the data are available from the authors upon reasonable requests.

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

## Acknowledgements

We thank to Joo-Von Kim and Jacques Miltat for critical reading of the manuscript. This work has been supported by the Agence Nationale de la Recherche (France) under Contract No. ANR-14-CE26-0012 (Ultrasky), ANR-09-NANO-002 (Hyfont), the RTRA Triangle de la Physique (Multivap), the European Research Council (ERC-StG-2014, Imagine) and by the Government of the Russian Federation, (Grant 074-U01).

## Author contributions

S.R., A.H. and A.T. conceived the study. A.H. and S.R. deposited and characterized the multilayer samples. A.H. and R.W. patterned the samples. A.H. performed the MFM measurements. S.R., J.S. and A.T. performed the micromagnetic simulations. M.B. participated in performing the BLS measurements and analysed all the BLS data. All authors discussed the data and reviewed the manuscript.

## Additional information

**Competing interests:** The authors declare no competing financial interests.

**Publisher's note**: 

