## [Peer Review File · Nature Communications]

Reviewers' comments:

Reviewer #1 (Remarks to the Author):

This manuscript discusses the stabilization and motion of magnetic skyrmions in wires patterned out of thin films composed of ferromagnetic and heavy metal layers, the later to promote DMI interaction and allow for the motion of the magnetic skyrmions through the Spin Hall effect. The main novelty of the work lies in the design of the system, where they demonstrate that by using two sets of FM/HM layers stacked oppositely around a space, dipolar fields can be used to enhance the chirality and stabilize coupled magnetic skyrmions. This implies that even for systems with weaker DMI, by design, a system can be created that will stabilize magnetic skyrmions. The current from a tipped electrode is shown nucleate skyrmions, and the gyrotropic motion of the skyrmions is demonstrated, evidencing a topological $S=1$ state of the skyrmions.

This is a nice paper that would be of interest to the community working on magnetic skyrmions in thin films, so I can recommend publication. However the following concern should be dealt with.

- Regarding the gyrotropic motion, it is calculated that the deflection angle should be about 71 degrees, and stated that this value is close to what is experimentally observed. However, the experimental data presented does not show such a deflection angle. Figure 2c shows an angle of at most 25 degrees. The authors should clarify this point, or include data which shows the experimental result.

Reviewer #2 (Remarks to the Author):

The manuscript by Hrabec et al. demonstrates an interesting variant of dipolar coupled skyrmions in a magnetic heterostructure consisting of conventional Ni/Co/Ni ferromagnets (FM). The authors engineer two layers that are mirror images of each other, namely Pt/FM, and FM/Pt, and separated by a gold spacer. The symmetry of the design produces coupled skyrmions with opposite chirality on opposite layers. When a current flows through the platinum, opposite current polarizations are injected into opposing FM layers via the spin-Hall effect. Due to the reverse chirality these neighbouring FM layers, the skyrmions experience the same force.

The authors make use of magnetic force microscopy to demonstrate that the skyrmion pairs flow along micron-wide wires in lock step with the application of large current densities. They have designed nano-contacts to the skyrmion material and demonstrate skyrmion injection into the heterostructure.

While the design is novel, the results of the manuscript are incremental. Skyrmion nucleation via a localized current injection from an STM-tip was demonstrated by Romming et al. in Ref. 7. Other groups have observed the skew deflection of skyrmions. More importantly, Woo et al. recently observed the transport of skyrmions via the spin-Hall effect in similar Pt/CoFe and Pt/Co/Ni multilayers by in Ref. 11. In these multilayers, the skyrmions are also coupled together by dipolar forces. The critical currents are a little lower, and the wall velocities are a little higher in the Pt/CoFe multilayers. Therefore, there is no clear benefit to the Pt/FM/Au/FM/Pt structure.

Although the manuscript is well written, and the idea is of interest, I cannot recommend that it be published in Nature Communications. In my opinion, the manuscript would be more suited to Scientific Reports.

Reviewer #3 (Remarks to the Author):

Report on „Current-induced skyrmion generation and dynamics in symmetric bilayers“ by A. Hrabec et al.

In their work, the authors report an experimental study of skyrmion stabilization in symmetric metallic bilayers due to dipolar and Dzyaloshinskii-Moriya interactions as well as current-induced creation and motion of skyrmions. The mechanism of skyrmion stabilization is explained based on micromagnetic simulations. The bilayers are composed of Pt/Ni/Co/Ni/Au(d)/Ni/Co/Ni/Pt stacks, the magnetic structure in external magnetic field is imaged by magnetic force microscopy (MFM) and transport measurements were performed which show creation and current-induced skyrmion motion and confirm the topological nature of the localized spin structures. The topic of this work is very interesting and timely and the idea of this approach is novel making this manuscript in principle suitable for publication in Nature Communications. However, I have a number of issues which the authors need to resolve before I can recommend publication.

The authors discuss stabilization of two skyrmions in the two ferromagnetic layers of the bilayer and argue that the dipolar interaction can be used to achieve the coupling. However, it is not mentioned in the manuscript, e.g. in the introduction, that the interlayer exchange coupling has also been proposed for this purpose and in how far it contributes in the present system in which the Au thickness is only 3 nm. There is one short note that the thickness has been optimized to avoid interlayer exchange coupling but there is no explanation just a reference which is very unsatisfactory.

From the MFM images there is no clear understanding of the internal spin structure of the localized spin structures which are denoted as skyrmions based on transport measurements. There is one short note that the diameter is about 300 nm, however, it is not clear whether these spin structures are skyrmions in the strict sense or rather skyrmionic bubbles in which the magnetization rotates only within chiral domain walls surrounding the core. This point needs to be clarified and clearly stated in the paper – also in the abstract and introduction.

The stabilization mechanism via dipolar interactions is based on the discussion of domains and domain walls. The micromagnetic calculations – as far as I understand the manuscript – do not explicitly take a skyrmion spin structure into account. However, from my point of view this is necessary in order to confirm the validity of this mechanism for skyrmions. In particular, it is important whether the energy considerations apply only for skyrmionic bubbles or also for skyrmions in which the magnetization rotates continuously from the core to the ferromagnetic surrounding.

In the manuscript the authors argue that the reported dipolar stabilization mechanism of skyrmions in bilayers is of great importance for new technologies based on skyrmions. However, it is not clear in how far this stabilization mechanism can be used also for small size skyrmions of diameters from 10 to 50 nm. The reported system shows relatively large skyrmionic spin structures with diameters of about 300 nm.

Detailed response to the reviewers:

Reviewer #1 :

This manuscript discusses the stabilization and motion of magnetic skyrmions in wires patterned out of thin films composed of ferromagnetic and heavy metal layers, the later to promote DMI interaction and allow for the motion of the magnetic skyrmions through the Spin Hall effect. The main novelty of the work lies in the design of the system, where they demonstrate that by using two sets of FM/HM layers staked oppositely around a space, dipolar fields can used to enhance the chirality and stabilized coupled magnetic skyrmions. This implies that even for systems with weaker DMI, by design, a system can be created that will stabilize magnetic skyrmions. The current from a tipped electrode is shown nucleate skyrmions, and the gyrotropic motion of the skyrmions is demonstrate, evidencing a topological $S=1$

state of the skyrmions.

This is a nice paper that would be of interest to the community working on magnetic skyrmions in thin films, so I can recommend publication. However the following concern should be dealt with.

- Regarding the gyrotropic motion, it is calculated that the deflection angle should be about 71 degrees, and stated that this value is close to what is experimentally observed. However, the experimental data presented does not show such a deflection angle. Figure 2c shows an angle of at most 25 degrees. The authors should clarify this point, or include data which shows the experimental result.

We agree with the referee that our strong statement here lacks more evidence. We didn't intend to exactly quantify the skyrmion Hall angle because it is clear that disorder plays an important part in our system and the entire interpretation is rather complex. As we explain below, a quantitative agreement can hardly be obtained in our system, the deflection is clearly demonstrated, with a sign in agreement with expectations.

- The referee refers to the angle deduced from Fig.2c which indeed corresponds to about 25 degrees. However, in this case, with a 1 μm wide track and given a skyrmion size of 160 nm, the edges strongly limit the deflection adding a repulsive force from the edge (such a force originates from canted spins at the track edges). This is why for a quantitative agreement, wider tracks should be considered. This is the purpose of the 3 μm wide track. However, the large deflection angle here (due to a small skyrmion size) implies that skyrmions still reach quite fast the track edges. From Fig.3b, one could infer a deflection angle of about 45°, considering a line connecting the nucleation electrode and the four aligned skyrmions on its right. In Fig.3c, skyrmions are found even behind the nucleation electrode, having been pushed by the large number of nucleated skyrmions via skyrmion-skyrmion interaction, so that the angle cannot be determined.
- A second limitation to the model is the important role of defects. It is now widely accepted that the skyrmion Hall angle in disordered systems depends on the magnitude of the driving force. This has been witnessed by experimental works (Jiang et al. Nature Physics 2016, Litzius et al. Nature Physics 2016) as well as by numerical calculations (Reichardt et al. Phys. Rev. Lett. 2015, Kim et al arXiv (2017)). From all our results (in particular Fig.2) it is clear that we didn't reach velocities sufficiently large to avoid an influence of the defects. This probably explains why the expected deflection angle is larger than the one that could be estimated from our experiments.

The manuscript has been corrected as follows:

- The strong statement "which is close to the experimental evidence" has been removed.
- The skyrmion size has been actualized to 160+/-40nm diameter (previous value was that in full films) since it is this size which governs the dynamics.
- The note concerning gyrotropic deflection and the related analytical calculations in the Supplementary Materials has been updated.
- References to Litzius et al. Nature Physics 2016 and Reichardt et al. Phys. Rev. Lett. 114, 217202 (2015), Kim et al arXiv: 1701.08357 (2017) have been added.

Reviewer #2 :

The manuscript by Hrabec et al. demonstrate an interesting variant of dipolar coupled skyrmions in a magnetic heterostructure consisting of conventional Ni/Co/Ni ferromagnets (FM). The authors engineer two layers that are mirror images of each other, namely Pt/FM, and FM/Pt, and separated by a gold spacer. The symmetry of the design produces coupled skyrmions with opposite chirality on opposite layers. When a current flows through the platinum, opposite current polarizations are injected into opposing FM layers via the spin-Hall effect. The due to the reverse chirality these neighbouring FM layers, the skyrmions experience the same force.

The authors make use of magnetic force microscopy to demonstrate that the skyrmion pairs flow along micron-wide wires in lock step with the application of large current densities. They have designed nano-contacts to the skyrmion material and demonstrate skyrmion injection into the heterostructure.

While the design is novel, the results of the manuscript are incremental. Skyrmion nucleation via a localized current injection from an STM-tip was demonstrated by Romming et al. in Ref. 7. Other groups have observed the skew deflection of skyrmions. More importantly, Woo et al. recently observed the transport of skyrmions via the spin-Hall effect in similar Pt/CoFe and Pt/Co/Ni multilayers by in Ref. 11. In these multilayers, the skyrmions are also coupled together by dipolar forces. The critical currents are a little lower, and the wall velocities are a little higher in the Pt/CoFe multilayers. Therefore, there is no clear benefit to the Pt/FM/Au/FM/Pt structure.

Although the manuscript is well written, and the idea is of interest, I cannot recommend that it be published in Nature Communications. In my opinion, the manuscript would be more suited to Scientific Reports.

We thank the referee for their positive comments about the technical quality of the manuscript. We would like to take the opportunity to allay the concerns raised about the novelty of our work.

As the referee mentions, all the achievements presented here are not new but the sample structure is novel. As explained in the introduction of the manuscript, the development of skyrmion-based functionality relies on three main aspects: skyrmion stabilization at room temperature, skyrmion nucleation and skyrmion efficient motion. Among these three points, the two last ones need to be independent. Separately, these aspects have been already demonstrated in several papers. However, this study is the first to show all of them in a single device. Moreover, we demonstrate for the first time nucleation using a non-ferromagnetic point contact (in contrast with Romming et al. Science 2013, where a spin-polarized tip was used). Even more interestingly, we have shown that playing with a point contact on one side of the track and a flat contact on the other side, nucleation and motion could be investigated independently, which has never been realized before.

A second novel point in this manuscript relies on the sample structure, which the referee found to be novel. As explained in the introduction, all the studies so far have relied on a medium with strong DMI, which limits the choice of materials. Here, we have successfully shown a new perspective to combine DMI with dipolar couplings, which requires a globally symmetric sample. With a relatively weak DMI of 0.2 mJ/m^2 (compared to other studies using more than 2 mJ/m^2), we succeeded in stabilizing 160 nm diameter skyrmions. This study will allow future studies to use a larger range of materials. According to the referee, dipolar couplings have already been used to stabilize skyrmions, in particular by Woo et al. We may add to this reference the work of Boulle et al. (Nature Nano 2016), who discusses such an effect in great detail. However, the situation is different here. In the previous studies, the dipolar coupling concerns the coupling of the skyrmion core to the uniform surrounding, involving the

perpendicular component of the field. In that case, it doesn't change the domain wall energy but increases the skyrmion stability. While in our case this effect is useful for the stabilization, a symmetric design also allows playing with the in-plane component of the stray field, coupling the domain wall magnetization to the skyrmion core and uniform surrounding. This effect, only possible in a bilayer with opposite chiralities in both layers is new and useful to avoid the need for a very large DMI. Our simulations (not shown) show that, in the absence of DMI, this effect could even be sufficient to stabilize skyrmions. We believe that with respect to the presented recent publications, our work meets the criterion of being "novel and important research study of high quality and of interest to that specific research community" for publication in Nature Communications, and hope that the referees agree with us.

The manuscript has been corrected as follows:

- We have added into the abstract a sentence that our bilayer system allows to lift the limitation of a large DMI for the skyrmion stabilization.

Reviewer #3:

Report on „Current-induced skyrmion generation and dynamics in symmetric bilayers“ by A. Hrabec et al.

In their work, the authors report an experimental study of skyrmion stabilization in symmetric metallic bilayers due to dipolar and Dzyaloshinskii-Moriya interactions as well as current-induced creation and motion of skyrmions. The mechanism of skyrmion stabilization is explained based on micromagnetic simulations. The bilayers are composed of Pt/Ni/Co/Ni/Au(d)/Ni/Co/Ni/Pt stacks, the magnetic structure in external magnetic field is imaged by magnetic force microscopy (MFM) and transport measurements were performed which show creation and current-induced skyrmion motion and confirm the topological nature of the localized spin structures. The topic of this work is very interesting and timely and the idea of this approach is novel making this manuscript in principle suitable for publication in Nature Communications. However, I have a number of issues which the authors need to resolve before I can recommend publication.

The authors discuss stabilization of two skyrmions in the two ferromagnetic layers of the bilayer and argue that the dipolar interaction can be used to achieve the coupling. However, it is not mentioned in the manuscript, e.g. in the introduction, that the interlayer exchange coupling has also been proposed for this purpose and in how far it contributes in the present system in which the Au thickness is only 3 nm. There is one short note that the thickness has been optimized to avoid interlayer exchange coupling but there is no explanation just a reference which is very unsatisfactory.

In order to experimentally verify that we deal with a system with a negligible exchange coupling, we have grown an additional stack composed of Pt\FM(t_1)\Au(3nm)\FM(t_2)\Pt where the thickness t_1 is different from t_2 . The minor hysteresis loops shown in Supplementary Materials indicate that the exchange coupling is negligible. This is consistent with the cited literature.

From the MFM images there is no clear understanding of the internal spin structure of the localized spin structures which are denoted as skyrmions based on transport measurements. There is one short note that the diameter is about 300 nm, however, it is not clear whether these spin structures are skyrmions in the strict sense or rather skyrmionic bubbles in which the magnetization rotates only within chiral domain walls surrounding the core. This point needs to be clarified and clearly stated in the paper – also in the abstract and introduction.

The referee asks if the observed textures are really skyrmions or simply skyrmionic bubbles. This underlines the difficulty of defining accurately what a skyrmion is and how a difference can be made between skyrmions and bubbles. Skyrmions and bubbles have in common the fact that they are localized textures (small domains). Bubbles in usual media (i.e. non chiral), as the ones used in bubble memories, can have topologies different from the trivial $S=0$ topology (in this case, they include two Bloch lines), $S=1$ (with no Bloch line), $S=-1$ (same exchange energy as $S=1$) or even much larger S number. The $S=1$ bubbles can be called skyrmionic bubbles, the $S=-1$ can be called antiskyrmionic bubbles. In this case, the textures cannot be called skyrmion as the energy difference between $S=0$ or $S=\pm 1$ bubbles is tiny (the cost of a Bloch line is small and even negligible for very large bubbles). In a chiral medium as the one considered in our study, the domain wall chirality is fixed and the cost of a Bloch line is too large so that all textures can only be $S=1$, as proved by our transport measurements (sensitivity to the spin Hall effect, gyrotropic deflection). The definition of a skyrmion in recent literature is not limited by its size. In the study by Jiang *et al.* (Nature Physics 2016) the observed skyrmions are a few μm large and are still called skyrmions (the magneto-optical images clearly show that the core is collinear). In the paper by Boulle *et al.* (Nature Nano 2016) the PEEM images also clearly show that the skyrmions have a collinear core for a 130 nm diameter skyrmion. In the paper by Romming *et al.* PRL 114 177203 (2015) the 6 nm wide skyrmions also have a collinear core as the domain wall width is much smaller than the skyrmion size. In the pioneering paper of Bogdanov and Hubert (JMMM 138, 255 (1994)) on skyrmions, solutions showing either continuous rotation or collinear core followed by a rotation by a domain wall like rotation are displayed and calculated using the same formalism. Most of the community agrees that all of these textures are skyrmions. According to the referee, a skyrmion (“in the strict sense”) should display a continuous rotation of the spins from the core to the outside. It is not the case in our study (as well as in the references cited above, and in many other studies): a large fraction of the core is rather collinear and the rotation occurs in the surroundings, like in bubbles. However, we respectfully disagree with referee 3 concerning the definition. This point has been already addressed in one of our previous paper (Rohart and Thiaville PRB 2013) where a continuous transition has been calculated from ultrasmall skyrmions (also called arrow shaped skyrmions, that the referee would call “skyrmions in a strict sense”) to large skyrmions (also called bubble shaped skyrmions, with a significant fraction of the core with a rather homogeneous magnetization). In this paper, the same calculation leads to both shapes without any transition between both of them, the only parameter controlling the shape being the DMI, which governs the skyrmion size. The absence of transition underlines the fact that both textures are of the same nature. From the micromagnetic theory, the main

length scale of the problem is the domain wall width which fixes the length scale for the rotation of spins from up to down orientation. It is impossible that the magnetization rotates slower than this length scale. This explains why for large skyrmions, the shape consists of a collinear core and a transition similar to a domain wall and that in ultrasmall skyrmions, the transition is faster, imposed by the DMI, which leads to an arrow shape. Similarly to our theory, the paper of Bogdanov and Hubert, where both shapes are also displayed, clearly shows that in the case of large anisotropies (ie. small domain wall width), the bubble-like skyrmion solution is found.

The stabilization mechanism via dipolar interactions is based on the discussion of domains and domain walls. The micromagnetic calculations – as far as I understand the manuscript – do not explicitly take a skyrmion spin structure into account. However, from my point of view this is necessary in order to confirm the validity of this mechanism for skyrmions. In particular, it is important whether the energy considerations apply only for skyrmionic bubbles or also for skyrmions in which the magnetization rotates continuously from the core to the ferromagnetic surrounding.

In order to address this issue we have performed additional complex micromagnetic simulations not only for skyrmions but also for 360° domain walls as an intermediate case. The outcome of these simulations is shown in Figs.S4 and S5 for 360° domain walls and skyrmions respectively. These results are consistent with the approach based on simple magnetic domain walls demonstrated in the main text and so prove the validity of this approach, a decreased strength of the flux closure effect being observed only below 20 nm domain wall separation in 360° walls.

We note here that while performing the simulations we have revealed a mistake in our parameters coming out from the chosen anisotropy. The chosen anisotropy (120mT), which slightly differs from the experimentally determined anisotropy (150mT), reproduces well the measured skyrmion size (160nm±40nm) from the Figs.2b and c. This now has been corrected across the entire paper.

In the manuscript the authors argue that the reported dipolar stabilization mechanism of skyrmions in bilayers is of great importance for new technologies based on skyrmions. However, it is not clear in how far this stabilization mechanism can be used also for small size skyrmions of diameters from 10 to 50 nm. The reported system shows relatively large skyrmionic spin structures with diameters of about 300 nm.

The above-mentioned simulations allowed us also to look into this problem in a greater detail. It is shown that the dipolar mechanism in this particular case is efficient for skyrmions diameters down to 40nm. This work therefore provides the route to develop materials for dipolar stabilization of even smaller skyrmions. We thank the referee for this excellent remark which allowed us to deepen the understanding of the underlying stabilizing mechanism.

The manuscript has been corrected as follows:

- We have mentioned in the main text that the exchange coupling has been proposed and also experimentally used to stabilize the skyrmion phase. We have added the experimental proof of negligible exchange coupling in our films into Supplementary Materials.

- We have added a new chapter into Supplementary Materials to show the modification of the flux closure mechanism from isolated domain walls to 360° domain walls and skyrmions down to 20nm diameter size. We have also added a phrase into the main text clarifying this point.

REVIEWERS' COMMENTS:

Reviewer #1 (Remarks to the Author):

In my opinion the authors of "Current-induced skyrmion generation and dynamics in symmetric bilayers" have fully and satisfactorily addressed my comments to the original paper, as well as those made by the other referees. In my opinion the paper's results are sufficiently novel that it warrant publication in Nature Communication, and I believe it will be well received by the community.

Reviewer #3 (Remarks to the Author):

The authors have clarified all the issues which I have raised in my previous report and I recommend publication of the manuscript in Nature Communications. In particular, I find the additional micromagnetic simulations of the new supplementary section II very important for the paper.